# Regional variation in brain tissue texture in patients with tonic-clonic seizures

Jennifer A. Ogren[1], Luke A. Allen[2], Bhaswati Roy[3], Beate Diehl[2], John M. Stern[4], Dawn S. Eliashiv[4], Samden D. Lhatoo[5], Ronald M. Harper[1,6], Rajesh Kumar[3,6,7,8]*

1 Department of Neurobiology, University of California at Los Angeles, Los Angeles, California, United States of America, 2 Department of Clinical and Experimental Epilepsy, University College London Institute of Neurology, London, United Kingdom, 3 Department of Anesthesiology and Perioperative Medicine, University of California Los Angeles, Los Angeles, California, United States of America, 4 Department of Neurology, University of California Los Angeles, Los Angeles, California, United States of America, 5 Department of Neurology, McGovern Medical School, University of Texas Health Science Center at Houston, Houston, Texas, United States of America, 6 Brain Research Institute, University of California Los Angeles, Los Angeles, California, United States of America, 7 Department of Radiological Sciences, University of California Los Angeles, Los Angeles, California, United States of America, 8 Department of Bioengineering, University of California Los Angeles, Los Angeles, California, United States of America

* rkumar@mednet.ucla.edu

**Data Availability Statement:** Since the data contain potentially sensitive personal health information on individual subjects, we are bound by ethical and legal restrictions on freely sharing

## Abstract

Patients with epilepsy, who later succumb to sudden unexpected death, show altered brain tissue volumes in selected regions. It is unclear whether the alterations in brain tissue volume represent changes in neurons or glial properties, since volumetric procedures have limited sensitivity to assess the source of volume changes (*e.g.*, neuronal loss or glial cell swelling). We assessed a measure, entropy, which can determine tissue homogeneity by evaluating tissue randomness, and thus, shows tissue integrity; the measure is easily calculated from T1-weighted images. T1-weighted images were collected with a 3.0-Tesla MRI from 53 patients with tonic-clonic (TC) seizures and 53 healthy controls; images were bias-corrected, entropy maps calculated, normalized to a common space, smoothed, and compared between groups (TC patients and controls using ANCOVA; covariates, age and sex; SPM12, family-wise error correction for multiple comparisons, p<0.01). Decreased entropy, indicative of increased tissue homogeneity, appeared in major autonomic (ventromedial prefrontal cortex, hippocampus, dorsal and ventral medulla, deep cerebellar nuclei), motor (sensory and motor cortex), or both motor and autonomic regulatory sites (basal-ganglia, ventral-basal cerebellum), and external surfaces of the pons. The anterior and posterior thalamus and midbrain also showed entropy declines. Only a few isolated regions showed increased entropy. Among the spared autonomic regions was the anterior cingulate and anterior insula; the posterior insula and cingulate were, however, affected. The entropy alterations overlapped areas of tissue changes found earlier with volumetric measures, but were more extensive, and indicate widespread injury to tissue within critical autonomic and breathing regulatory areas, as well as prominent damage to more-rostral sites that exert influences on both breathing and cardiovascular regulation. The entropy measures provide easily-collected supplementary information using only T1-weighted images, showing aspects of tissue integrity other than volume change that are important for assessing function.

data publicly. However, data from this study are available with approval from the UCLA Institutional Review Board for researchers who meet the criteria for access to confidential data. This restriction is due to the consent used at the time of data collection. Data may be requested by contacting the UCLA Institutional Review Board (Website: https://webirb.research.ucla.edu/; Phone (Medical IRB): +1-310-825-5344; Email: webirbhelp@research.ucla.edu; Reference: IRB#14-001301.

**Funding:** This work was supported by the National Institute of Neurologic Disorders and Stroke [U01 NS090407]. The UCL group is grateful to the Wolfson Foundation and the Epilepsy Society for supporting the Epilepsy Society MRI scanner. The UCL contributions were also supported by the National Institute for Health Research, University College London Hospitals Biomedical Research Centre. The funders had no role in study design, data collection and analysis, decision to publish, or preparation of the manuscript.

**Competing interests:** The authors have declared that no competing interests exist.

# 1. Introduction

A concern in patients with epilepsy is the potential for progressive brain injury in critical structures from repeated seizures, which may partially result from impaired blood perfusion arising out of extreme activation of autonomic nervous system influences on the vasculature during seizures. Excitotoxic damage from hyperactivity of neuronal processes, which is another potential mechanism for tissue injury, may be compounded by inadequate perfusion to sustain underlying metabolic needs of overactive neurons and supporting cells. Such processes impose more a concern if the affected brain structures mediate functions essential for survival, which include both maintenance of blood pressure and ventilation. The consequences of repeated injury are apparent; Sudden Unexpected Death in Epilepsy (SUDEP) accounts for ~7.5–17% of deaths in individuals with epilepsy, and over 50% of deaths in patients with intractable epilepsy [1]. Patients with Generalized or Bilateral Tonic-Clonic seizures (TCs) are at elevated risk for SUDEP, with seizure frequency related to that risk [2, 3], a most-probable outcome of enhanced brain injury with successive ictal events. Although the precise mechanisms contributing to SUDEP remain unclear, the processes likely involve a combination of respiratory and/or cardiovascular dysfunction, with sustained apnea leading to reduced oxygenation, cardiac arrhythmia, or profound hypotension [4]. The potential for such events would be exacerbated if breathing and cardiovascular regulatory brain structures, or normal recovery mechanisms from oxygen desaturation or hypotension were impaired.

Evaluation of the functional relevance of injured brain areas would be simplified if the affected areas uniformly lost volume, with declining volumes in those regions translating to impaired function. However, overall-brain evaluation by T1-weighted imaging-based volumetric procedures, although typically showing decreased volumes, sometimes substantially, also show enhanced volumes. Cortical areas in patients with TCs show localized brain tissue thinning, but also thickening, with regional thickness frequently varying in sites affecting cardiovascular or breathing action [5]. Patients with epilepsy at variable risk for SUDEP show brain areas mediating breathing and cardiovascular action undergo volume loss in several regions, but volume gain in other regions. The volume losses, as determined by T1-weighted volumetric procedures in subsequent victims of SUDEP, are often extreme in areas essential for recovery from prolonged apnea and resulting hypoxia or hypotension [6, 7], but increased volumes in other sites important for instigating apnea or collapsing blood pressure are also present. Thus, procedures for assessing competence of regional tissue function based on volumetric determination alone appear insufficient.

Volumetric tissue changes and measures of cortical thickness, determined by T1-weighted procedures, have limited capability to assess the nature of those increased or decreased tissue volume changes. Such measures are unable to differentiate between neuronal loss from glial or other tissue changes with inflammation, due to the restricted range of values on gray matter volume assessment and inherent spatial resolution issues. These limitations make insights into the nature and extent of brain tissue alterations difficult.

Tissue integrity can, however, be examined with texture assessments using procedures derived from high-resolution T1-weighted images, and these texture measures are sensitive to the nature and extent of tissue changes. Tissue texture measures quantify patterns of image signal intensities that differ with the variable nature and extent of brain tissue changes. Although there are several tissue-texture measures, entropy is one such evaluation that assesses the extent of homogeneity or randomness of tissue water signals based on signal intensity characteristics. Entropy is based on information theory, which is translated to a biological system, and shows data complexity, with complex data requiring more data points to characterize and indicate higher entropy. Highly isotropic tissues, such as free water in cerebrospinal fluid,

show a greater number of equal states, and fewer data are required to describe simple tissue, leading to lower entropy [8, 9]. However, in the case of highly organized brain tissues, such as white matter sites, due to the complex nature more information is required to accurately describe the tissue, and thus, entropy values will be higher. Entropy values are inversely proportional to the amount of free water content within tissue, with reduced extracellular water corresponding to cellular and axonal swelling. The entropy values can thus help clarify the processes underlying volumetric tissue changes, with increased and decreased regional volumes, found earlier in high-risk SUDEP patients [6, 10, 11].

Entropy measures are useful to determine regional tissue changes in homogeneity; more-homogenous texture is associated with acute injury severity [12]. The potential for entropy values to reveal acute vs chronic tissue alterations, and to provide insights into tissue swelling vs tissue loss may help resolve mechanisms underlying the volume changes found in earlier volumetric studies of patients with epilepsy who are at risk for SUDEP. Assessment of tissue texture over the entire brain could elucidate the temporal sequencing of mechanisms contributing to brain tissue changes that lead to altered breathing or cardiovascular patterns over time in these patients at risk.

A practical aspect of tissue integrity evaluation by entropy measures bears consideration. Tissue entropy can be assessed using simple calculations on conventional T1-weighted scans. Although tissue integrity can be estimated with diffusion tensor imaging scans [13–15], such data may not always be collected in patients with epilepsy, while T1-weighted image acquisitions are common in all clinical sites. The potential exists for reduced costs for current evaluation of tissue changes in patients with epilepsy, as well as for post-hoc evaluation of existing T1-weighted data.

We examined tissue texture patterns indicative of brain changes in TCs vs healthy subjects. We hypothesized that, compared to healthy controls, brain entropy values would be altered in TCs patients, especially in cardiovascular and respiratory regions, and provide insights into the nature of tissue injury in patients at risk for SUDEP.

## 2. Materials and methods

### 2.1 Subjects

Data from 53 patients with TCS (mean age ± SD: 36.6±12.6 years, 23 male, 43 right-handed) were collected from sites at Case Western Reserve, University College, London, and University of California at Los Angeles (UCLA). Epilepsy patients admitted at Epilepsy Monitoring Units at Case Western Reserve, University College, London, and UCLA sites were approached to discuss the study and whether they wished to participate. All epilepsy patients showed a high incidence of generalized TCS and were at significant risk for SUDEP. Data from 53 healthy control subjects (mean age ± SD: 37.3±14.6 years, 23 male, 34 right-handed) were acquired at the University of California Los Angeles. Subject demographics are outlined in Table 1. The study was approved by the Institutional Review Board at the University of California at Los Angeles, and informed written consent was obtained from each participant before data collection.

### 2.2 Magnetic resonance imaging

Brain studies at UCLA and Case Western Reserve were performed using a 3.0-Tesla MRI scanner (Siemens, Magnetom, Erlangen, Germany). Side foam pads were used bilaterally to avoid head motion, and subjects laid supine during the scanning. High-resolution T1-weighted images were collected using a magnetization prepared rapid acquisition gradient-echo

**Table 1. Demographics and clinical characteristics of GTCS patients and control subjects.**

| Variables | GTCS | Controls | P values |
|---|---|---|---|
| | n = 53 | n = 53 | |
| | (Mean ± SD) | (Mean ± SD) | |
| Age (years) | 36.6±12.6 | 37.3±14.6 | 0.80 |
| Sex [male] (%) | 23 (43%) | 23 (43%) | 1.00 |
| BMI | (n = 44) 27.2 ± 5.6 | 24.2 ± 4.4 | 0.003 |
| Handedness [L/R/ambidex] | [4/47/2] | [12/34/7] | 0.01 |
| Disease duration (years) | 15.8 ± 12.2 | - | - |
| GTCS/months | (n = 43) 6.2 ± 27.2 | - | - |
| Versive (Y/N) | (n = 52) 15/ 37 | - | - |
| Nocturnal (Y/N) | 32/ 21 | - | - |

GTCS, generalized tonic-clonic seizure; SD = standard deviation; BMI = body mass index; Y = yes; N = no.

sequence (TR = 2200 ms; TE = 3.05 ms; inversion time = 1100 ms; FA = 10˚; matrix size = 256×256; FOV = 220×220 mm$^2$; slice thickness = 1.0 mm; slices = 176).

## 2.3 Data processing and analysis

Multiple software packages were used for image visualization, data pre-processing, and analyses, and included the Statistical Parametric Mapping package SPM12 (Wellcome Department of Cognitive Neurology, UK; http://www.fil.ion.ucl.ac.uk/spm), MRIcroN (https://www.nitrc.org), and MATLAB-based custom routines (The MathWorks Inc, Natick, MA). High-resolution T1-weighted images of all subjects were visually-examined to ensure that no serious brain pathologies (e.g., cyst, tumor, or infarct) were present before data processing, and none of the controls included here showed any serious brain tissue changes.

## 2.4 Entropy calculation

High-resolution T1-weighted images were bias-corrected to remove any signal intensity variations due to field inhomogeneities with SPM12 software. Using the bias-corrected T1-weighted images, the entropy values at a given voxel 'v' were calculated with the following equation by defining a 3×3×3 volume of interest (VOI) with 'v' as center:

$$E = -\sum_{i=1}^{N} p_i \log(p_i)$$

Where, N is number of distinct pixel values (gray/white matter) in the VOI, and pi is the probability of occurrence of i$^{th}$ pixel value in the VOI. Mathematically, the VOI can be represented as below:

$$VOI = I(x - 1.5 : x + 1.5, y - 1.5 : y + 1.5, z - 1.5 : z + 1.5)$$

where, 'I' is the bias corrected T1-weighted image, and x, y, z are spatial coordinates.

## 2.5 Normalization and smoothing of entropy maps

Before normalization of entropy maps to a common space, the entropy values were scaled between 0–1 by dividing the whole-brain entropy by its maximum value, to attain a common distribution of values across the different scanners. Whole-brain entropy maps were then normalized to Montreal Neurological Institute (MNI) space using the SPM12 package. The

warping parameters for x, y, z directions were obtained from the bias-corrected T1-weighted images via modified unified segmentation approach, and resulting parameters were applied to the corresponding entropy maps. The normalized entropy maps were smoothed using an iso-tropic Gaussian filter (8 mm kernel).

### 2.6 Statistical analyses

The SPM12 and the IBM statistical package for social sciences (IBM SPSS v26, Armonk, New York) were employed for statistical analyses. Chi-square and independent-samples t-tests were used to examine group differences in demographic and other clinical data. We considered a p-value less than 0.05 as statistically significant.

The smoothed entropy maps were compared voxel-by-voxel between groups using analysis of covariance (ANCOVA), with age and sex included as covariates (SPM12, family-wise error correction for multiple comparisons, p<0.01). The global brain mask was used to restrict the analysis within brain regions only, and sites with significant differences between groups were overlaid onto background images for structural identification.

## 3. Results

### 3.1 Regions with increased homogeneity in GTC patients

Decreased entropy values, indicative of increased homogeneity, appeared prominently in areas associated with blood pressure and breathing control, extending from the ventral medial fron-tal cortex (Fig 1h) through the basal forebrain and sub anterior cingulate areas, and hypothala-mus (Fig 1d), anterior and posterior thalamus (Fig 1g), midbrain (Fig 1f), external surface of the pons (Fig 1a), dorsal and ventral medulla (Fig 1e), deep cerebellar nuclei (Fig 1b) and ven-tral basal cerebellum Fig 1c).

In addition to ventral medial prefrontal cortex, (Fig 1h), sensorimotor cortices (Fig 1l), pos-terior insula (Fig 2), ventral temporal cortices (Fig 1m), and posterior cingulate (Fig 1i) showed entropy declines. A large portion of the anterior pole of the temporal lobe showed lower entropy values, and included the hippocampus and amygdala (Figs 1n and 2a and 2b). The basal ganglia (Fig 2d) sites were severely affected bilaterally. Portions of the thalamus (Fig 1g), except for deep areas (Fig 1j), and the midbrain (Fig 1f) were affected, as well as the pons, dorsal and ventral medulla, and ventral-basal cerebellum (Fig 1a, 1c and 1e). The pons and thalamus showed unique entropy distributions in that the external surfaces showed reduced entropy values, but not deeper tissue (Figs 1j and 1k and 2c). Among the spared autonomic control regions was the anterior cingulate and anterior insula; both the posterior insula and cingulate were, however, affected.

### 3.2 Sites with reduced homogeneity in GTC patients

Only a few isolated brain regions showed increased entropy, reflecting decreased tissue homo-geneity, resulting from long-term injury. These sites included areas within the occipital cortex.

## 4. Discussion

Alterations in regional brain cortical thicknesses, earlier described in patients with TC sei-zures, are accompanied by changes in entropy, indicating disrupted underlying structural organization of the affected tissue. Decreased entropy (increased tissue homogeneity) appeared in multiple areas classically defined as sensory or motor (sensory and motor cortex, basal ganglia, cerebellum), regions that serve motor, cognitive, memory or affective functions, in addition to autonomic regulation (basal ganglia [16–18], deep autonomic cerebellar nuclei

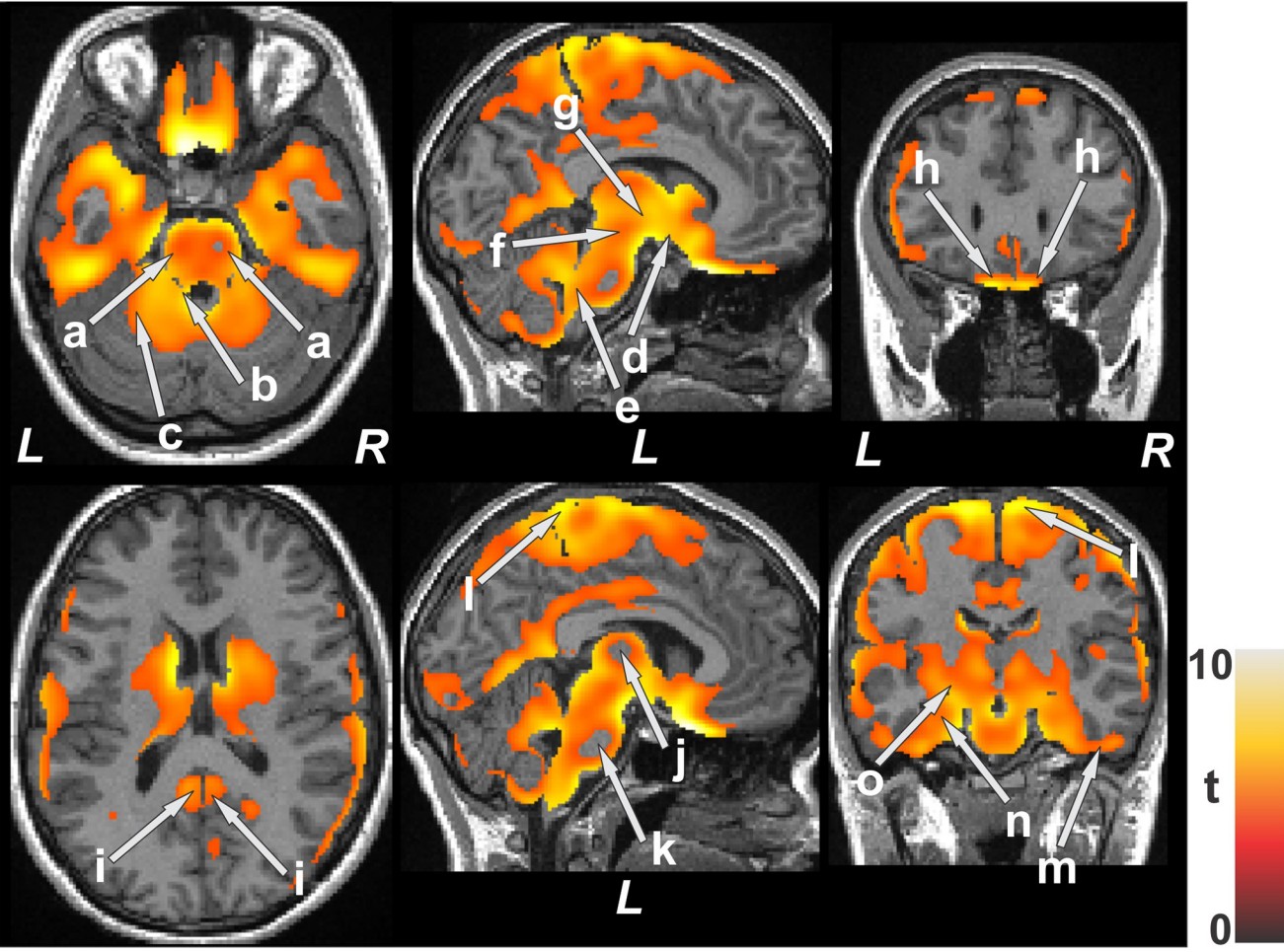

**Fig 1. Brain areas demonstrating decreased entropy (colored) in dorsal and ventral pons (a), deep cerebellar nuclei (b), anterior cerebellum (c), ventral medial frontal cortex (h), extending through the hypothalamus and basal forebrain (d) to the thalamus (g), midbrain (f), dorsal and ventral medulla (e) in TC patients vs controls (p<0.01), corrected for multiple comparisons.** Near-midline slice illustrating extensive entropy declines bilaterally in the ventral temporal cortex (m), and includes the hippocampus (n) and the basal ganglia (o); sparing of deeper tissue in the thalamus (j) and pons (k) is apparent. Extensive entropy declines bilaterally in the sensorimotor cortex (l), as well as posterior cingulate (i) in TC patients appeared.

and cerebellar cortex [19–21], ventral medial prefrontal [22–25], anterior pole and ventral temporal cortices, hypothalamus, posterior cingulate, insula, amygdala, hippocampus, and dorsal and ventral medulla, suggesting that acute TC seizures establish tissue alterations which could induce long-term dysfunction in sites serving multiple traditional functions, as well as autonomic and respiratory regulation.

Of particular concern for reports of sustained central apnea in the post-ictal period is the decreased entropy in the rostral and ventral temporal lobe encompassing the amygdala and portions of the hippocampus. The amygdala, with its direct projections to the periaqueductal gray, a structure that provides excitation to breathing, and to respiratory phase-switching para-brachial pontine areas [26], can play a significant role in eliciting prolonged apnea. Single-pulse stimulation to the central amygdala in awake feline preparations can trigger inspiratory efforts [27], but train stimulation elicits apnea in pediatric patients [28], and amygdala lesions reduce seizure-induced respiratory arrest in DBA/1 mice [29].

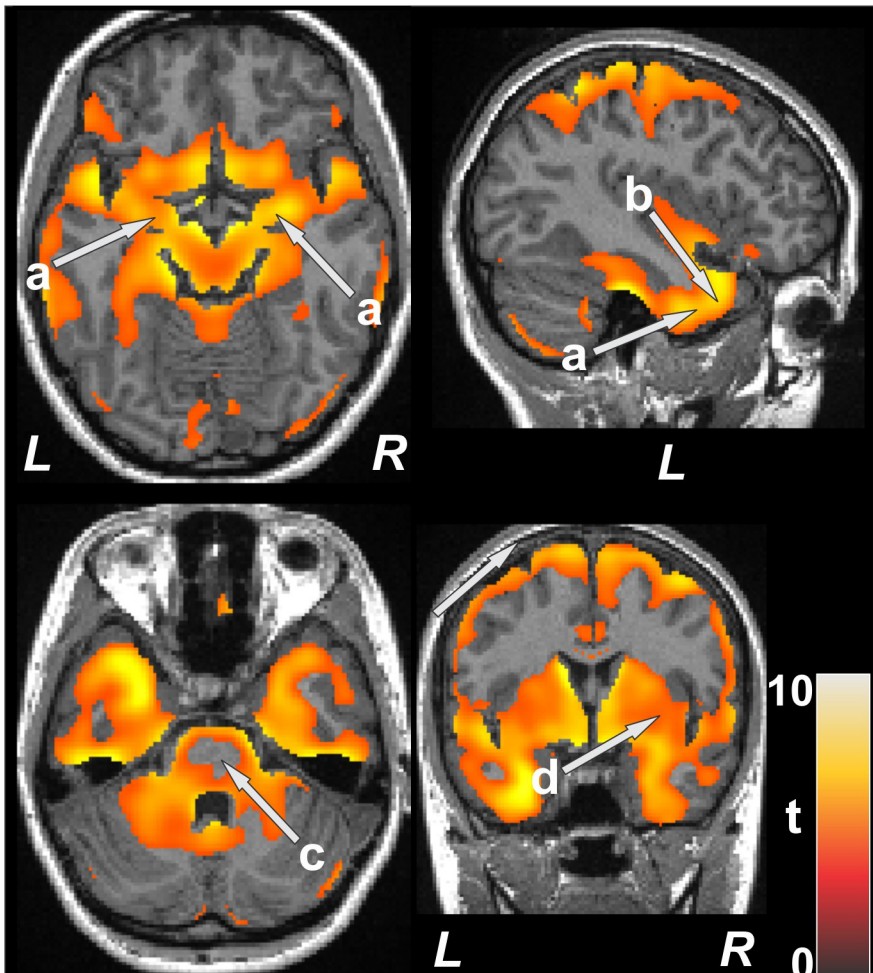

**Fig 2. Amygdala (a), temporal lobe (b), and basal ganglia (d) showed decreased entropy values in TC patients vs controls (p<0.01).** Few changes in deep regions of the pons (c) appear, but reduced entropy is found on the surface.

The hippocampus is traditionally associated with memory and related cognitive functions. However, the hippocampus serves significant blood pressure and breathing roles; the structure is a vital link in rostral brain regulation of blood pressure [25] that responds prominently to multiple blood pressure challenges in humans during functional MRI [25]. Moreover, regional hippocampal single neuron discharge follows the breathing cycle in humans [30], and train electrical stimulation of the ventral hippocampus in the rat induces apnea, slows subsequent respiratory rate, and increases tidal volume [31]. The blood pressure effects can be abolished by ventral medial frontal cortex lesions; that area shows significant entropy declines here. Nearly the entire anterior and ventral temporal lobe showed decreased entropy, indicating tissue structural changes potentially affecting a range of temporal cortex-related functions.

Similarly, brain sites normally associated with motor coordination, such as the cerebellum, (showing significant entropy changes in this study), are critical in recovery from prolonged apnea [20] and profound blood pressure collapse [19], roles that are important for SUDEP. We found significant entropy changes in the basal ganglia, normally considered an extrapyramidal motor control area. However, recent evidence, largely derived from studies of Parkinson's disease, indicates pronounced basal ganglia roles in autonomic regulation, especially for

parasympathetic action [18], and thus, critical in mediating bradycardia sometimes found preceding SUDEP.

Actions of the sensorimotor and ventral cerebellar regions control a range of respiratory musculature ranging from the diaphragm, abdominal, and thoracic muscles, as well as upper airway flow-restricting muscles, could be compromised given the injury found here. Although selected autonomic sites were spared (anterior insula, anterior cingulate), the ventral medial prefrontal cortex, cerebellar deep nuclei, and medulla showed significant entropy changes, which, if transformed into chronic injury has the potential to impede resting and dynamic changes to blood pressure.

Few areas of increased entropy (decreased homogeneity), presumably reflecting the enhanced structural organization accompanying the increased connectivity found in particular pathways in those who succumb to SUDEP [32] were found. These areas included portions of the occipital cortex, and the processes underlying those entropy changes are unclear. Such patterns of little common injury are unlike those of temporal lobe epilepsy onset patients, where increased entropy appears in multiple, defined sites.

We speculate that those outcomes reflect enhanced vascularization of tissue on the structural surfaces relative to deeper tissues, with well-perfused areas more affected by perfusion changes accompanying ictal events in these two sites. Such a possibility has implications for protection of brain areas during ictal events. If the speculation is indeed the case, efforts at reducing constriction of the central vasculature, perhaps by restricting extremes of central hypertension during ictal events might lessen central injury. Such an objective might be achieved by prophylactic pharmacologic agents, or by recently described neuromodulatory procedures, such as been found useful for apnea in premature infants [33].

Declines in entropy typically reflect a loss in tissue organization, a process found with inflammation, which often accompanies acute tissue changes. Earlier T1-weigthed imaging-based cortical thickness studies [5] showed regions of both decreased and increased cortical thickness over controls; these areas of cortical thinning included the frontal cortex, temporal pole, posterior cingulate and lateral parietal cortices, and those sites appeared here as regions of reduced entropy, i.e., decline in tissue organization. Increased cortical thickness has been reported in post-central gyri, insula, and subgenual, anterior, posterior, and isthmus cingulate cortices [5]. The latter finding, increased cortical thickness which, in the current study, showed decreased entropy, suggests the possibility of inflammatory processes (*i.e.*, reduced organization, not enhanced neural tissue development in those cortical areas). Analogous tissue changes can be found in some disease conditions, e.g., acute hepatic encephalopathy, which shows increased regional brain volumes, reflected as increased cortical thickness accompanying inflammatory processes.

Perhaps the findings of most interest for SUDEP investigators are the indications of decreased entropy in classic areas of cardiovascular and respiratory patterning influence. These sites include the temporal pole and amygdala, the ventral medial prefrontal cortex, the hippocampus, and anterior and posterior thalamus, dorsal and ventral midbrain, the entire medullary area, and projections to the deep cerebellar nuclei. All of these areas influence the cardiovascular system or respiratory timing in significant ways. The amygdala is the focus of processes that trigger apnea or respiratory rate through projections to the periaqueductal gray and parabrachial pons [26], and functional demonstrations by stimulation [28] or lesion evidence [29]. The ventral medial prefrontal cortex and hippocampal role in mediation of blood pressure has been well described [25], the posterior thalamus serves roles in mediating hypoxia and expiratory timing [34, 35], and has shown tissue loss earlier [10], both dorsal and ventral medullary areas are critical in both chemo and lung afferent sensing for breathing and blood

pressure, and the cerebellar deep nuclei play a significant role in compensating for extremes of both blood pressure loss or prolonged apnea [19, 20].

Entropy procedures have been useful in assessing neural injury in other clinical conditions. Pediatric obstructive sleep apnea (OSA) is accompanied by a wide range of tissue injury to the brain, a consequence of intermittent hypoxia exposure, and those injuries vary in severity, depending on brain area. Detectability of gray matter volume changes in pediatric OSA is often difficult, given the short time course of disease development over, for example, adult OSA, the latter often taking years to emerge. T1-weighted scans and gray matter volume analysis alone are inadequate to detect tissue type changes in pediatric OSA cases, but entropy procedures were much more useful [36].

Other evaluations of tissue integrity in epilepsy typically use variations of diffusion tensor imaging (DTI) to evaluate the motion of water molecules within tissue as a function of direction to reveal the underlying tissue microstructure. Extensions of DTI techniques can reveal the packing density of neurites and the spatial organization of the neuronal projections [37]. However, a significant consideration in a clinical environment is the potential to derive a measure of tissue integrity rapidly and at low cost. The entropy measures can be calculated easily from T1-weighted scans that are typically acquired from patients with epilepsy in all clinical sites, while other assessments of damage, e.g., DTI, may not be available. The need for only T1-weighted images also makes available retrospective examination of tissue integrity for data collected before DTI procedures were commonly used.

## 5. Limitations

One limitation of this study includes unavailability of data and analyses from different types of seizures instead of a defined cohort. However, other types of seizures should also show entropy changes, but the sites and direction of entropy changes (increased or decreased) may vary, based on acute or chronic tissue changes in different types of epilepsy patients. Another major limitation of this study is that the entropy values were determined solely from adults; pediatric cases with epilepsy would require a very large sample with snapshots of structure at different developmental stages of changes in neural tissue and glia accompanying the normal rapid transitions of such tissue in young children. Pediatric cases offer the possibility of following the interaction of injurious intermittent hypoxia and impaired perfusion on tissue homogeneity accompanying ictal events on differing cortical and other tissue sites with development; cortical changes alone during early teen age years are substantial [for Review, see [38]]. However, such an evaluation was outside of the scope of this study, and we studied only adult subjects. Similarly, the sample size was insufficient to adequately assess duration of epilepsy in subjects or proximity to the last seizure.

The patients evaluated were subjected to a wide array of antiepileptic medications. The action of these agents on both glial and neuronal cellular processes have not been well-described, but could have affected the properties of tissue here, just as pharmacologic agents affect cortical tissue thickness [39].

We cannot assume that the increased or decreased entropy values resulted from unitary processes; it is possible that a combination of pathologic changes could have led to assessed values that were less deviant from normal levels.

## 6. Conclusions

Brain areas critical for mediating a range of cognitive, affective, and motor control show increased homogeneity, i.e., a decline in differentiation of tissue in epilepsy patients with TCS. Several of these areas serve primary respiratory and cardiovascular functions; however,

multiple regions shared both respiratory and cardiovascular roles with motor, memory, and affective functions, and these areas were widespread. The affected regions typically overlapped sites of previously described volume loss and increases, determined by T1-weighted imaging procedures, and likely resulted from inflammatory processes. Well-defined structures, such as the thalamus and pons show surface entropy declines with spared deeper tissue, possibly from ictal influences on the surface vasculature of these structures. The anterior cingulate and anterior insula were also spared. Entropy evaluation can provide insights into the nature of injury in brain structures of patients with epilepsy and do so using only T1-weighted imaging.

## Acknowledgments

We thank Ms. Rebecca K. Harper for assistance with data collection.

## Author Contributions

**Conceptualization:** Ronald M. Harper, Rajesh Kumar.

**Funding acquisition:** Beate Diehl, Samden D. Lhatoo, Ronald M. Harper, Rajesh Kumar.

**Investigation:** Jennifer A. Ogren, Luke A. Allen.

**Methodology:** Luke A. Allen, Bhaswati Roy.

**Resources:** Luke A. Allen, Beate Diehl, Samden D. Lhatoo.

**Supervision:** Rajesh Kumar.

**Validation:** Rajesh Kumar.

**Writing – original draft:** Jennifer A. Ogren, Rajesh Kumar.

**Writing – review & editing:** Luke A. Allen, Bhaswati Roy, John M. Stern, Dawn S. Eliashiv, Samden D. Lhatoo, Ronald M. Harper, Rajesh Kumar.

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
