## [Decision Letter · Decision Letter 0]

31 May 2022

PONE-D-22-08022Regional Variation in Brain Tissue Texture in Patients with Tonic-Clonic SeizuresPLOS ONE

Dear Dr. Kumar,

Thank you for submitting your manuscript to PLOS ONE. After careful consideration, we feel that it has merit but does not fully meet PLOS ONE’s publication criteria as it currently stands. Therefore, we invite you to submit a revised version of the manuscript that addresses the points raised during the review process.

The manuscript makes a new contribution to the literature by showing the effects of entropy changes following tonic-clonic seizures. Detailed critiques of the manuscript by the two reviewers are presented below. Please respond to each of the critiques of the reviewers. A selection of the primary comments are summarized below.

1. More details regarding recruitment of patients and their histories would be helpful, including their etiologies.

2. Further discussion and analysis of entropy, including how it could be used clinically.

3. Possible limitations/concerns regarding the use of entropy should be considered, including the lack of specificity (which should be discussed and which appears to be prevalent from the observed data).

4. Further analysis of other forms of epilepsy as they may relate to and affect entropy.

We look forward to receiving your revised manuscript.

Kind regards,

Allan Siegel

Academic Editor

PLOS ONE

Journal Requirements:

“This work was supported by the National Institute of Neurologic Disorders and Stroke [U01 NS090407]. The UCL group is grateful to the Wolfson Foundation and the Epilepsy Society for supporting the Epilepsy Society MRI scanner. The UCL contributions were also supported by the National Institute for Health Research, University College London Hospitals Biomedical Research Centre.”

“This work was supported by the National Institute of Neurologic Disorders and Stroke [U01 NS090407]. The UCL group is grateful to the Wolfson Foundation and the Epilepsy Society for supporting the Epilepsy Society MRI scanner. The UCL contributions were also supported by the National Institute for Health Research, University College London Hospitals Biomedical Research Centre.

Additional Editor Comments:

The manuscript makes a new contribution to the literature by showing the effects of entropy changes following tonic-clonic seizures. Detailed critiques of the manuscript by the two reviewers are presented below. Please respond to each of the critiques of the reviewers. A selection of the primary comments are summarized below.

1. More details regarding recruitment of patients and their histories would be helpful, including their etiologies.

2. Further discussion and analysis of entropy, including how it could be used clinically.

3. Possible limitations/concerns regarding the use of entropy should be considered, including the lack of specificity (which should be discussed and which appears to be prevalent from the observed data).

4. Further analysis of other forms of epilepsy as they may relate to and affect entropy.

Reviewers' comments:

Reviewer's Responses to Questions

**Comments to the Author**

1. Is the manuscript technically sound, and do the data support the conclusions?

Reviewer #1: Partly

Reviewer #2: Partly

2. Has the statistical analysis been performed appropriately and rigorously? 

Reviewer #1: Yes

Reviewer #2: Yes

3. Have the authors made all data underlying the findings in their manuscript fully available?

Reviewer #1: Yes

Reviewer #2: Yes

4. Is the manuscript presented in an intelligible fashion and written in standard English?

Reviewer #1: Yes

Reviewer #2: Yes

5. Review Comments to the Author

Reviewer #1: In this paper, authors present data on entropy measures in patients with tonic-clonic (TC) seizures and healthy controls. The present evidence of decreased entropy (increased tissue homogeneity) in major autonomic regulatory areas, motor control areas and the pons which are widespread. They concluded that injury to areas of the brain associated with control of breathing and circulation occurs due to TC seizures and may be the mechanism underlying SUDEP. The study was well conducted and imaging data presented was interesting. I also think this paper adds to the literature on changes in brain (volumetric) in patients with or at risk of SUDEP.

There are several issues that are not clear and would benefit or enhance the impact.

1. They recruited patients with TC seizures. However, addition of several pieces of data would be helpful. a. How did they determine that these patients are at risk of SUDEP? Is there a risk score (Hesdoffer et al, 2011 for example)? b. what types of epilepsy patients are we dealing with? c. no demographic data is presented. I would recommend addition of patient details and a method to classify those that have TC seizures but at "higher risk of SUDEP" vs low risk of SUDEP".

2. The entropy changes are wide spread and not limited to finite regions of the brain. Not all of these areas are associated with breathing and circulation. In fact, the changes in entropy are seen in areas far beyond and hence the interpretation that "entropy changes are seen in areas that regulate breathing and circulation" may not be accurate. It is possible that they recruited patients with many different etiologies and a wide age spectrum that may have resulted in this. Once they brain in all patient data (in the form of 1-2 tables), it may be more clear and conclusions can be appropriately adjusted.

3. Lines 194-199 is a long run on sentence and confusing to read. Please modify or simplify.

Reviewer #2: This study sought to identify the effects of clonic-tonic seizures upon neuronal cell tissue volume within different regions of the brain. The study is of interest and the findings and methodology could be of potential importance. However, a number of issues are raised in the manuscript that the authors should address. These are indicated below.

To what extent are the changes in entropy unique to tonic-clonic seizures? Or would such effects also be seen with other kinds of seizures such as partial-complex seizures that do not evolve into generalized tonic-clonic seizures or myoclonic seizures, etc.? Further analysis of other types of seizures would have been helpful here.

It is not clear the extent to which entropy can be used as a clinical measure or estimate of possible neurological damage or dysfunction. Do the authors have any evidence from other know and well-established methodologies and tools that correlate with entropy as a neurological tool or measure? The question raised here is to what extent does the change in entropy reflect a clinical phenomenon.

Another issue is the extent of the specificity of the effects of seizures as determined by entropy measures. For example, the authors describe a wide number of structures displaying changes in entropy. But how many of these are related to cardiovascular changes that possibly are linked to sudden death syndrome? A specific example: in Fig, 2, changes are shown in entropy for components of the basal ganglia…but this region is generally not known to be associated with cardiovascular events. So, how do the authors know which structures are relevant to the processes possibly linked to the cardiovascular and related effects of the tonic-clonic seizures?

Fig. 1 indicates the effects of seizures upon entropy`/ What about the brains of normal individuals (controls)?

In the beginning of the manuscript, the authors raised the question of whether entropy was related to neurons or glia. Do the authors have any thoughts on this question?

6. PLOS authors have the option to publish the peer review history of their article (what does this mean?). If published, this will include your full peer review and any attached files.

Reviewer #1: No

Reviewer #2: **Yes: **Allan Siegel

---

## [Author Response · Author response to Decision Letter 0]

11 Aug 2022

Response to Reviewers

PONE-D-22-08022

Regional Variation in Brain Tissue Texture in Patients with Tonic-Clonic Seizures

PLOS ONE

Dear Dr. Kumar,

Thank you for submitting your manuscript to PLOS ONE. After careful consideration, we feel that it has merit but does not fully meet PLOS ONE’s publication criteria as it currently stands. Therefore, we invite you to submit a revised version of the manuscript that addresses the points raised during the review process. The manuscript makes a new contribution to the literature by showing the effects of entropy changes following tonic-clonic seizures. Detailed critiques of the manuscript by the two reviewers are presented below. Please respond to each of the critiques of the reviewers. A selection of the primary comments are summarized below.

1. More details regarding recruitment of patients and their histories would be helpful, including their etiologies.

We appreciate this suggestion. We expanded subject recruitment details, patient histories, and added etiologies for available subjects.

2. Further discussion and analysis of entropy, including how it could be used clinically.

Yes. We understand this concern. We added a paragraph in the Discussion on clinical usefulness and prior studies under other medical conditions, as well as advantageous practical considerations of the entropy measures. A significant consideration is the potential to derive a measure of tissue integrity solely from T1-weighted scans, typically acquired from patients with epilepsy in all clinical centers, where other assessments of damage, e.g., diffusion tensor imaging, may not be available. That potential offers considerable cost benefits, as well as the possibility to evaluate tissue integrity retrospectively in the absence of DTI data. Those aspects are now discussed as well.

3. Possible limitations/concerns regarding the use of entropy should be considered, including the lack of specificity (which should be discussed, and which appears to be prevalent from the observed data).

The specificity issue is indeed a concern. This aspect is now discussed in the context of the widespread tissue injury accompany epilepsy in diverse brain areas, as well as in the limitations of implementation. We also consider the lack of specificity in the context of the central concern for SUDEP, namely injury to cardiovascular and breathing regulatory sites, and discuss how multiple “motor” and “cognitive” sites serve autonomic and breathing roles (thus, spreading functions over multiple brain areas).

4. Further analysis of other forms of epilepsy as they may relate to and affect entropy.

We agree that adding analyses of other forms of epilepsy would be helpful to generalize the findings, and to provide insights into the nature of brain tissue injury. However, we lack availability of such data and would not be able to add additional analyses in the paper. We do point to another study in pediatric obstructive sleep apnea, a condition with both marked and subtle brain tissue injury, where entropy measures were successfully used.

Done!

Uploaded.

Done.

There are no changes in the financial disclosures from the original submission.

Guidelines for resubmitting your figure files are available below the reviewer comments at the end of this letter. If applicable, we recommend that you deposit your laboratory protocols in protocols.io to enhance the reproducibility of your results. Protocols.io assigns your protocol its own identifier (DOI) so that it can be cited independently in the future. For instructions see: https://journals.plos.org/plosone/s/submission-guidelines#loc-laboratory-protocols. Additionally, PLOS ONE offers an option for publishing peer-reviewed Lab Protocol articles, which describe protocols hosted on protocols.io. Read more information on sharing protocols at https://plos.org/protocols?utm_medium=editorial-email&utm_source=authorletters&utm_campaign=protocols.

Not applicable.

We look forward to receiving your revised manuscript.

Kind regards,

Allan Siegel

Academic Editor

PLOS ONE

 Journal Requirements:

Checked!

Not applicable.

“This work was supported by the National Institute of Neurologic Disorders and Stroke [U01 NS090407]. The UCL group is grateful to the Wolfson Foundation and the Epilepsy Society for supporting the Epilepsy Society MRI scanner. The UCL contributions were also supported by the National Institute for Health Research, University College London Hospitals Biomedical Research Centre.” We note that you have provided additional information within the Acknowledgements Section that is not currently declared in your Funding Statement. Please note that funding information should not appear in the Acknowledgments section or other areas of your manuscript. We will only publish funding information present in the Funding Statement section of the online submission form. Please remove any funding-related text from the manuscript and let us know how you would like to update your Funding Statement. 

We removed the funding-related information from the manuscript per the journal’s requirement.

Currently, your Funding Statement reads as follows: “This work was supported by the National Institute of Neurologic Disorders and Stroke [U01 NS090407]. The UCL group is grateful to the Wolfson Foundation and the Epilepsy Society for supporting the Epilepsy Society MRI scanner. The UCL contributions were also supported by the National Institute for Health Research, University College London Hospitals Biomedical Research Centre. The funders had no role in study design, data collection and analysis, decision to publish, or preparation of the manuscript.” Please include your amended statements within your cover letter; we will change the online submission form on your behalf.

The amended Funding Statement included in the cover letter that reads as follows: “This work was supported by the National Institute of Neurologic Disorders and Stroke [U01 NS090407]. The UCL group is grateful to the Wolfson Foundation and the Epilepsy Society for supporting the Epilepsy Society MRI scanner. The UCL contributions were also supported by the National Institute for Health Research and University College London Hospitals Biomedical Research Centre. The funders had no role in study design, data collection and analysis, decision to publish, or preparation of the manuscript.”

4. We note that you have indicated that data from this study are available upon request. PLOS only allows data to be available upon request if there are legal or ethical restrictions on sharing data publicly. For more information on unacceptable data access restrictions, please see http://journals.plos.org/plosone/s/data-availability#loc-unacceptable-data-access-restrictions. In your revised cover letter, please address the following prompts:

The cover letter is revised and details the ethical and legal restrictions on freely sharing data publicly. However, such data will be shared with investigators upon request, as long as personal identifying information is not needed for their analyses. 

b) If there are no restrictions, please upload the minimal anonymized data set necessary to replicate your study findings as either Supporting Information files or to a stable, public repository and provide us with the relevant URLs, DOIs, or accession numbers. For a list of acceptable repositories, please see http://journals.plos.org/plosone/s/data-availability#loc-recommended-repositories. We will update your Data Availability statement on your behalf to reflect the information you provide.

ORCID ID is available for the corresponding author, and this ID is validated in Editorial Manager.

Reviewers' comments:

Reviewer's Responses to Questions

Comments to the Author

1. Is the manuscript technically sound, and do the data support the conclusions? The manuscript must describe a technically sound piece of scientific research with data that supports the conclusions. Experiments must have been conducted rigorously, with appropriate controls, replication, and sample sizes. The conclusions must be drawn appropriately based on the data presented. 

Reviewer #1: Partly

Reviewer #2: Partly

2. Has the statistical analysis been performed appropriately and rigorously? 

Reviewer #1: Yes

Reviewer #2: Yes

3. Have the authors made all data underlying the findings in their manuscript fully available? 

Reviewer #1: Yes

Reviewer #2: Yes

4. Is the manuscript presented in an intelligible fashion and written in standard English? PLOS ONE does not copyedit accepted manuscripts, so the language in submitted articles must be clear, correct, and unambiguous. Any typographical or grammatical errors should be corrected at revision, so please note any specific errors here.

Reviewer #1: Yes

Reviewer #2: Yes

5. Review Comments to the Author

Reviewer #1: In this paper, authors present data on entropy measures in patients with tonic-clonic (TC) seizures and healthy controls. The present evidence of decreased entropy (increased tissue homogeneity) in major autonomic regulatory areas, motor control areas and the pons which are widespread. They concluded that injury to areas of the brain associated with control of breathing and circulation occurs due to TC seizures and may be the mechanism underlying SUDEP. The study was well conducted and imaging data presented was interesting. I also think this paper adds to the literature on changes in brain (volumetric) in patients with or at risk of SUDEP.

There are several issues that are not clear and would benefit or enhance the impact.

1. They recruited patients with TC seizures. However, addition of several pieces of data would be helpful. 

a. How did they determine that these patients are at risk of SUDEP? Is there a risk score (Hesdoffer et al, 2011 for example)?

We appreciate the reviewer’s query. We did not evaluate epilepsy patients for low or high-risk of SUDEP category. However, a characteristic of our patient data set was a high incidence of generalized tonic clonic seizures (GTCS), which, as the reviewer notes, is a significant risk factor for SUDEP. We clarified this issue in the manuscript. 

b. what types of epilepsy patients are we dealing with?

All patients included in this study expressed GTCS with variable seizure frequency. 

c. no demographic data is presented. 

Added. Thank you!

I would recommend addition of patient details and a method to classify those that have TC seizures but at "higher risk of SUDEP" vs low risk of SUDEP".

Thank you for the suggestion. We added patient details, including recruitment information, demographics, and other available clinical data. In addition, seizure frequency details that correlate with SUDEP risk based on previous studies from others are added, although SUDEP risk classification information is not included, since such classifications were not performed in this study. 

2. The entropy changes are wide spread and not limited to finite regions of the brain. Not all of these areas are associated with breathing and circulation. In fact, the changes in entropy are seen in areas far beyond and hence the interpretation that "entropy changes are seen in areas that regulate breathing and circulation" may not be accurate. It is possible that they recruited patients with many different etiologies and a wide age spectrum that may have resulted in this. Once they brain in all patient data (in the form of 1-2 tables), it may be more clear and conclusions can be appropriately adjusted.

We included epilepsy patients with consistent etiologies, with a relatively narrow age range. In addition, age was included as a covariate in the statistical model. We now realize that we did not adequately describe how brain areas, not typically considered “respiratory” or “cardiovascular” control sites, may influence such vital functions (and showed entropy changes). Our findings indicate significant tissue changes in autonomic and respiratory regulatory areas, as well as in sites usually associated with motor, mood and cognitive functions. However, it is now apparent that brain areas known to be primarily associated with negative emotions, such as the amygdala, play a key role in inducing apnea in human patients with epilepsy [1-3], and can pace breathing in animal models [4]. Similarly, brain sites normally associated with motor coordination, such as the cerebellum (showing significant entropy changes in this study), are critical in recovery from prolonged apnea and profound blood pressure collapse [5, 6], outcomes that are important for SUDEP. We found significant entropy changes in the basal ganglia, normally considered an extrapyramidal motor control area. However, recent evidence, largely derived from studies of Parkinson’s disease, indicates pronounced basal ganglia roles in autonomic regulation, especially for parasympathetic action, and thus, critical in bradycardia sometimes found preceding SUDEP.

The potential for cerebellar and amygdala dysfunctions to play key roles in breathing and blood pressure is now being recognized in the epilepsy field as a principal area of concern in SUDEP [7-10]. However, other brain sites normally associated with mood or cognitive functions are now realized to have essential breathing or cardiovascular roles, and some of these sites showed dramatic entropy changes. These areas include the cingulate cortex (serving mood and depression; but also blood pressure [11, 12], hippocampus (memory, but a key structure in blood pressure regulation [13, 14], ventral medial prefrontal cortex (significant structure in central apnea, as well as blood pressure) [15-17], and midbrain (essential breathing and blood pressure roles) [15, 16]. We realize that those descriptions were lacking in the original manuscript, and those aspects are now included. 

3. Lines 194-199 is a long run on sentence and confusing to read. Please modify or simplify.

Modified (currently lines 270-274). Thank you!

Reviewer #2: This study sought to identify the effects of clonic-tonic seizures upon neuronal cell tissue volume within different regions of the brain. The study is of interest and the findings and methodology could be of potential importance. However, a number of issues are raised in the manuscript that the authors should address. These are indicated below.

To what extent are the changes in entropy unique to tonic-clonic seizures? Or would such effects also be seen with other kinds of seizures such as partial-complex seizures that do not evolve into generalized tonic-clonic seizures or myoclonic seizures, etc.? Further analysis of other types of seizures would have been helpful here. 

We believe that other types of seizures will also show entropy changes, but the direction of entropy changes (increased or decreased) may vary, based on acute or chronic tissue changes in different types of epilepsy patients. Our data are derived from a defined cohort, and we are unable to generalize the findings here to other types of seizures. 

It is not clear the extent to which entropy can be used as a clinical measure or estimate of possible neurological damage or dysfunction. Do the authors have any evidence from other known and well-established methodologies and tools that correlate with entropy as a neurological tool or measure? The question raised here is to what extent does the change in entropy reflect a clinical phenomenon.

That question is very appropriate. It is the case that the measure is novel, but has been used earlier to assess brain injury in pediatric obstructive sleep apnea (OSA) [18], a condition where T1-weighted scans and gray matter volume analysis alone are inadequate to detect tissue type changes. It should be noted that the injury detected by entropy measures in pediatric OSA cases has been confirmed by other MRI measures. One significant advantage of the entropy measure is the simplicity; only easy calculations from T1-weighted scans are needed, and T1-weighted scans are typically readily available for patients with epilepsy, while other tissue-change assessments, such as diffusion tensor imaging, may not be as readily attainable.

Another issue is the extent of the specificity of the effects of seizures as determined by entropy measures. For example, the authors describe a wide number of structures displaying changes in entropy. But how many of these are related to cardiovascular changes that possibly are linked to sudden death syndrome? A specific example: in Fig, 2, changes are shown in entropy for components of the basal ganglia…but this region is generally not known to be associated with cardiovascular events. So, how do the authors know which structures are relevant to the processes possibly linked to the cardiovascular and related effects of the tonic-clonic seizures?

This issue was partially covered in response to a concern raised by Reviewer 1. Several of the brain sites, normally considered as having few, or no cardiovascular or respiratory-related functions, in reality serve significant breathing or autonomic roles. The basal ganglia, long recognized as serving extrapyramidal motor control functions, serves significant autonomic functions, especially for the parasympathetic system; most of those revelations derived from studies of Parkinson’s Disease. Since bradycardia is a concern in SUDEP, integrity of the parasympathetic system is critical in patients with epilepsy. Although we emphasized that significant tissue changes occurred in autonomic and breathing control areas, we did not indicate that those areas were widespread, and included “non-conventional” sites, such as the basal ganglia. We showed tissue changes over the entire brain in the figures, and now point out the widely-distributed nature of brain sites that modify breathing and cardiovascular action. In addition, other cognitive and affective brain regions are affected as well. 

Fig. 1 indicates the effects of seizures upon entropy`/ What about the brains of normal individuals (controls)?

Figure 1 shows regional brain tissue entropy changes in patients with GTCS compared to normal/control subjects. Brain images of normal individuals were evaluated for any serious brain pathology before data processing, and none of the controls included here showed any serious brain tissue changes. We added this description in the manuscript. 

In the beginning of the manuscript, the authors raised the question of whether entropy was related to neurons or glia. Do the authors have any thoughts on this question?

We believe that entropy changes in this study result predominantly from glial changes, since the tissue alterations likely are indicative of inflammatory activities that contribute to glial cell activation. 

6. PLOS authors have the option to publish the peer review history of their article (what does this mean?). If published, this will include your full peer review and any attached files. 

Do you want your identity to be public for this peer review? For information about this choice, including consent withdrawal, please see our Privacy Policy.

Reviewer #1: No

Reviewer #2: Yes: Allan Siegel

References

1. Rhone AE, Kovach CK, Harmata GI, Sullivan AW, Tranel D, Ciliberto MA, et al. A human amygdala site that inhibits respiration and elicits apnea in pediatric epilepsy. JCI Insight. 2020;5(6). doi: 10.1172/jci.insight.134852. PubMed PMID: 32163374; PubMed Central PMCID: PMCPMC7213805.

2. Nobis WP, Schuele S, Templer JW, Zhou G, Lane G, Rosenow JM, et al. Amygdala-stimulation-induced apnea is attention and nasal-breathing dependent. Ann Neurol. 2018;83(3):460-71. doi: 10.1002/ana.25178. PubMed PMID: 29420859; PubMed Central PMCID: PMCPMC5867259.

3. Nobis WP, Gonzalez Otarula KA, Templer JW, Gerard EE, VanHaerents S, Lane G, et al. The effect of seizure spread to the amygdala on respiration and onset of ictal central apnea. J Neurosurg. 2019;132(5):1313-23. doi: 10.3171/2019.1.JNS183157. PubMed PMID: 30952127; PubMed Central PMCID: PMCPMC8022327.

4. Harper RM, Frysinger RC, Trelease RB, Marks JD. State-dependent alteration of respiratory cycle timing by stimulation of the central nucleus of the amygdala. Brain Res. 1984;306(1-2):1-8. doi: 10.1016/0006-8993(84)90350-0. PubMed PMID: 6466967.

5. Lutherer LO, Lutherer BC, Dormer KJ, Janssen HF, Barnes CD. Bilateral lesions of the fastigial nucleus prevent the recovery of blood pressure following hypotension induced by hemorrhage or administration of endotoxin. Brain Res. 1983;269(2):251-7. doi: 10.1016/0006-8993(83)90134-8. PubMed PMID: 6349747.

6. Xu F, Frazier DT. Role of the cerebellar deep nuclei in respiratory modulation. Cerebellum. 2002;1(1):35-40. doi: 10.1080/147342202753203078. PubMed PMID: 12879972.

7. Allen LA, Vos SB, Kumar R, Ogren JA, Harper RK, Winston GP, et al. Cerebellar, limbic, and midbrain volume alterations in sudden unexpected death in epilepsy. Epilepsia. 2019;60(4):718-29. doi: 10.1111/epi.14689. PubMed PMID: 30868560; PubMed Central PMCID: PMCPMC6479118.

8. Whatley BP, Winston JS, Allen LA, Vos SB, Jha A, Scott CA, et al. Distinct Patterns of Brain Metabolism in Patients at Risk of Sudden Unexpected Death in Epilepsy. Front Neurol. 2021;12:623358. doi: 10.3389/fneur.2021.623358. PubMed PMID: 34899550; PubMed Central PMCID: PMCPMC8651549.

9. La A, Rm H, M G, R K, Ja O, Sb V, et al. Altered brain connectivity in sudden unexpected death in epilepsy (SUDEP) revealed using resting-state fMRI. Neuroimage Clin. 2019;24:102060. doi: 10.1016/j.nicl.2019.102060. PubMed PMID: 31722289; PubMed Central PMCID: PMCPMC6849487.

10. Allen LA, Harper RM, Kumar R, Guye M, Ogren JA, Lhatoo SD, et al. Dysfunctional Brain Networking among Autonomic Regulatory Structures in Temporal Lobe Epilepsy Patients at High Risk of Sudden Unexpected Death in Epilepsy. Front Neurol. 2017;8:544. doi: 10.3389/fneur.2017.00544. PubMed PMID: 29085330; PubMed Central PMCID: PMCPMC5650686.

11. Gianaros PJ, Derbyshire SW, May JC, Siegle GJ, Gamalo MA, Jennings JR. Anterior cingulate activity correlates with blood pressure during stress. Psychophysiology. 2005;42(6):627-35. doi: 10.1111/j.1469-8986.2005.00366.x. PubMed PMID: 16364058; PubMed Central PMCID: PMCPMC2246096.

12. Burns SM, Wyss JM. The involvement of the anterior cingulate cortex in blood pressure control. Brain Res. 1985;340(1):71-7. doi: 10.1016/0006-8993(85)90774-7. PubMed PMID: 4027647.

13. Korf ES, White LR, Scheltens P, Launer LJ. Midlife blood pressure and the risk of hippocampal atrophy: the Honolulu Asia Aging Study. Hypertension. 2004;44(1):29-34. doi: 10.1161/01.HYP.0000132475.32317.bb. PubMed PMID: 15159381.

14. Feng R, Rolls ET, Cheng W, Feng J. Hypertension is associated with reduced hippocampal connectivity and impaired memory. EBioMedicine. 2020;61:103082. doi: 10.1016/j.ebiom.2020.103082. PubMed PMID: 33132184; PubMed Central PMCID: PMCPMC7585137.

15. Cechetto DF, Shoemaker JK. Functional neuroanatomy of autonomic regulation. Neuroimage. 2009;47(3):795-803. doi: 10.1016/j.neuroimage.2009.05.024. PubMed PMID: 19446637.

16. Shoemaker JK, Goswami R. Forebrain neurocircuitry associated with human reflex cardiovascular control. Front Physiol. 2015;6:240. doi: 10.3389/fphys.2015.00240. PubMed PMID: 26388780; PubMed Central PMCID: PMCPMC4555962.

17. Wong SW, Masse N, Kimmerly DS, Menon RS, Shoemaker JK. Ventral medial prefrontal cortex and cardiovagal control in conscious humans. Neuroimage. 2007;35(2):698-708. doi: 10.1016/j.neuroimage.2006.12.027. PubMed PMID: 17291781.

18. Kheirandish-Gozal L, Sahib AK, Macey PM, Philby MF, Gozal D, Kumar R. Regional brain tissue integrity in pediatric obstructive sleep apnea. Neurosci Lett. 2018;682:118-23. doi: 10.1016/j.neulet.2018.06.002. PubMed PMID: 29883682; PubMed Central PMCID: PMCPMC6102065.

---

## [Editor Report · Decision Letter 1]

30 Aug 2022

Regional Variation in Brain Tissue Texture in Patients with Tonic-Clonic Seizures

PONE-D-22-08022R1

Dear Dr.Kumar,

We’re pleased to inform you that your manuscript has been judged scientifically suitable for publication and will be formally accepted for publication once it meets all outstanding technical requirements.

Kind regards,

Allan Siegel

Academic Editor

PLOS ONE

Additional Editor Comments (optional):

In the initial review of the manuscript, the reviewers described four concerns with the original manuscript. The authors responded effectively to three of these concerns and the authors provided a reasonable explanation why they could not comply with the fourth issue. Overall, the manuscript makes a new, useful and interesting contribution to the literature and thus it is now at a level worthy of publication.
---

## [Editor Report · Acceptance letter]

14 Sep 2022

PONE-D-22-08022R1 

Regional Variation in Brain Tissue Texture in Patients with Tonic-Clonic Seizures 

Dear Dr. Kumar:

I'm pleased to inform you that your manuscript has been deemed suitable for publication in PLOS ONE. Congratulations! Your manuscript is now with our production department. 

Kind regards, 

on behalf of

Dr Allan Siegel 

Academic Editor

PLOS ONE